# Deep Learning Based Feature Selection Algorithm for Small Targets Based on mRMR

**DOI:** 10.3390/mi13101765

**Published:** 2022-10-18

**Authors:** Zhigang Ren, Guoquan Ren, Dinhai Wu

**Affiliations:** Department of Vehicle and Electrical Engineering, Shijiazhuang Branch, Army Engineering University of PLA, Shijiazhuang 050003, China

**Keywords:** feature extraction, minimum redundancy maximum relevance, deep learning network, image pre-processing

## Abstract

Small target features are difficult to distinguish and identify in an environment with complex backgrounds. The identification and extraction of multi-dimensional features have been realized due to the rapid development of deep learning, but there are still redundant relationships between features, reducing feature recognition accuracy. The YOLOv5 neural network is used in this paper to achieve preliminary feature extraction, and the minimum redundancy maximum relevance algorithm is used for the 512 candidate features extracted in the fully connected layer to perform de-redundancy processing on the features with high correlation, reducing the dimension of the feature set and making small target feature recognition a reality. Simultaneously, by pre-processing the image, the feature recognition of the pre-processed image can be improved. Simultaneously, by pre-processing the image, the feature recognition of the pre-processed image can significantly improve the recognition accuracy. The experimental results demonstrate that using the minimum redundancy maximum relevance algorithm can effectively reduce the feature dimension and identify small target features.

## 1. Introduction

The data volume of images, videos, and texts has grown exponentially due to the rapid development of neural networks and artificial intelligence. While the data volume is complex and multi-dimensional, there is a great deal of redundancy and irrelevant information [1]. How to deal with the curse of the dimensionality of useful features from data has become a research focus. Feature selection is a fundamental data dimensionality reduction technique. Extracting the feature sub-set with the best evaluation criteria from the original data and performing mathematical calculations on the original data, such as classification and regression, can significantly improve the computational efficiency of the algorithm and reduce learning errors. In short, using feature selection can reduce the feature dimension while increasing the learning model’s training speed. According to the number of labels in the samples, feature selection methods are classified as supervised, semi-supervised, or un-supervised [2,3,4]. Convolutional neural networks [5], for example, evaluate discriminative features based on sample labels and distinguish the importance of features based on the correlation between labels and features [6]. The feature is evaluated and selected in the un-supervised method by judging the correlation between the data structure and the feature [7,8]. The variance and Laplace scores are commonly used to assess the correlation between the features. It is difficult to select features in the un-supervised method, because there are no labels. Semi-supervised methods do not have a high labeling cost [9]. To generate pseudo-labels, researchers frequently use matrix decomposition, spectral clustering, and other methods [10]. Then combine sparse learning with feature selection. Semi-supervised learning achieves high performance with fewer labels and multiple samples, while also requiring less raw data by taking into account label information and structural information of all data. There are currently three main feature extraction algorithm processes. The first extracts the shape, position, and texture of grayscale target image features, before feeding the acquired feature data into the support vector machine classifier (SVM) for processing [11,12]. Classification quantifies image features using prior semantic information, models the quantized data, and then predicts. The third uses the rapid development of deep learning to initially calibrate the original data and then to learn the initial labels to predict the same type of features.

The supervised method’s convolutional neural network builds a feature detection and extraction model that can extract multiple features from the target image at the same time. This method can optimize the model’s search space and greatly improve the model’s convergence speed. Small and tiny targets have a poor learning ability. When the size and step size of the convolution kernel are reduced, the network is prone to local convergence, which eventually leads to the poor learning ability of small features, such as pneumonia and part defects [13,14]. The extraction of features is more difficult. There is a lack of labels for the research object categories in practical applications, and a large workload is required to discriminate the characteristics of different samples. Semi-supervised methods are, generally, only suitable for sample learning with no class differences. In order to address the aforementioned issues, this paper proposes a semi-supervised method for the feature selection of small targets based on the minimum redundancy maximum relevance feature selection algorithm. Redundant features are screened using the minimum redundancy maximum relevance method for each candidate feature [15,16,17].

In this paper, the deep learning method is used to extract features through comparative experiments, and the minimum redundancy maximum relevance (mRMR) algorithm is used for feature selection and extraction. This paper’s main contributions are as follows: the dependencies between the candidate features and the selected features are considered; the maximum correlation minimum redundancy is used to evaluate and select each feature; comparison experiments are used to accurately measure the models with and without the mRMR algorithm rate determination; it is demonstrated that the mRMR algorithm has a better choice for small target features by extracting pneumonia features and visualizing the extracted features.

## 2. Related Knowledge

### 2.1. Minimum Redundancy Maximum Relevance Algorithm

The minimum redundancy maximum relevance (mRMR) algorithm [9] works on the principle of selecting a set of features (Min-Redundancy) with the highest correlation of the final output result (Max-Relevance) and the lowest correlation between the features in the original feature set. Equation (1) is satisfied with the maximum correlation feature.
(1)$$\max D(S,c),D=\frac{1}{\left|S\right|}\sum_{x_{i}\in S}I(x_{i};c)$$

The feature sub-set S is comprised of m features $\left\{x_{i}\right\}$, and the feature is approximated with the average value D(S,c) of the mutual information between the classification targets c [18,19]. A significant amount of redundancy is generated between the features chosen by Max-Relevance, resulting in feature dependence. When two features are redundant, Min-Redundancy is used to eliminate the redundant features:(2)$$\min R(S),R=\frac{1}{{\left|S\right|}^{2}}\sum_{x_{i},x_{i}\in S}I(x_{i},x_{j})$$

By combining Max-Relevance with Min-Redundancy:(3)maxΦ(D,R),Φ=D−R

In practice, $S_{m-1}$ feature sub-sets are obtained, and m features must be chosen from the remaining $X-S_{m-1}$ sub-sets. If you wanted to maximize Φ(.), you would have to optimize the corresponding incremental algorithm. The following is the formula:(4)$$\underset{x_{j}\in X-S_{m-1}}{\max}[I(x_{j};c)-\frac{1}{m-1}\sum_{x_{i}\in S_{m-1}}I(x_{i};x_{j})]$$
where $I(x_{j};c)$ represents the *j*th characteristic in $X-S_{m-1}$.

### 2.2. Computer-Aided Diagnosis Using mRMR

Incorporating the mRMR algorithm into computer-aided diagnosis can reduce system memory usage and eliminate feature information with low correlation or redundancy [20,21]. In general, the computer-aided system can extract dozens, if not hundreds, of feature information from an image. A large portion of this data is redundant and useless. As a result, this paper selected the feature selection algorithm with the minimum redundancy maximum relevance to reduce the dimension of the extracted multi-dimensional features.

The system modules in this paper were mainly divided into data pre-processing, the purpose of which was to convert the acquired image files into .bmp format image files for easy processing. Image area segmentation, whose function was to narrow the search range of candidate features and remove feature parts other than those needed; candidate feature generation and feature extraction, the latter of which, in this paper, was based on the feature calculation formula of image science; improving the efficiency of program operation and the accuracy of classification; and training the SVM classifier [21], whose role was to build a classifier model for prediction. The overall architecture of the system is shown in Figure 1.

Each module operated in a serial fashion, and its functions were as follows:

Data-processing module: Various image formats were converted to a .bmp format image.

Image-preprocessing module: segmented the image using the image’s basic operations (erosion, filtering, flood filling, etc.).

Feature extraction module: located candidate features by clustering segmented images, then extracting features.

Feature selection module: using the mRMR method, it screened the extracted feature data and selected the feature set with the highest correlation with the category.

Classifier-training module: associated the filtered feature set with the category, generated training data, and trained the classifier.

Prediction module: used the obtained trainer model to determine predictions based on a given set of features.

There was also an unbreakable link between the various modules in the systems. The initial data set was pre-processed, the .bmp image was generated, and a feature extraction on the generated .bmp image was performed, resulting in a data set corresponding to the features and categories for training [22]. The data set contained many features that had little correlation with the category, and the feature selection module was used to filter the features. The filtered data set was saved after training the classifier. When the testing set was imported, the display module could load the classifier and perform a predictive diagnosis on the images.

### 2.3. Feature Selection Algorithm Proposed in This Paper

This paper performed a deep-learning-based feature extraction, employed the mRMR algorithm to filter all features in the final fully connected layer, and stored all acquired features as a data set. Data consisting of n instances are denoted as $Data=\left[inst_{1},\;inst_{2},\;\ldots ,\;inst_{i},\;\ldots ,\;inst_{n}\right]$; data denotes a data set, and $inst_{i}$ (1≤i≤n) denotes the i-th feature, which is represented as $inst_{i}=[f_{1}^{\left(i\right)},f_{2}^{\left(i\right)},\cdot \cdot \cdot ,f_{i}^{\left(i\right)},\cdot \cdot \cdot ,f_{m}^{\left(i\right)},C_{i}]$; $f_{j}^{i}$ is the feature vector of $inst_{i}$, and $C_{i}$ is the feature’s categorical variable.

Step 1. Define the candidate feature sub-set S and initialize *S* = ∅.

Step 2. Using the Max-Relevance method, the eigenvector group $F=[f_{1},f_{2},\cdots ,f_{i},\cdots ,f_{m}]$, and the category vector $C=[C_{1},C_{2},\cdots ,C_{i},\cdots ,C_{m}]$, calculate and record the correlation between each feature vector $f_{j}$ in F and the category vector C, in turn, denoted as $I\left(f_{j};C\right)$.

Step 3. Look for the feature vector with the Max-Relevance with the category vector C, and denote the feature vector as *S*_1_.

Step 4. Add $S_{1}$ to the candidate feature sub-set S, thereby obtaining an updated candidate feature sub-set $S^{\prime }$; delete $S_{1}$ from the feature vector group F, thereby obtaining a new feature vector group $F^{\prime }$.

Step 5. Assign the updated candidate feature sub-set $S^{\prime }$ to the candidate feature sub-set S; assign the updated feature vector group $F^{\prime }$ to F.

Step 6. Define the loop value k and initialize k = 2.

Step 7. According to the feature vector group F and the candidate feature sub-set S, use the mRMR to calculate the correlation redundancy value of each feature vector $f_{j}$ in F and S with respect to the category vector C, denoted as *J_s_*(*f_j_*).

Step 8. Find the vector with the largest correlation redundancy value with S in respect to the category vector C, and assign it to *S_k_*.

Step 9. Add $S_{k}$ to the candidate feature sub-set S to obtain an updated candidate feature sub-set $S^{\prime }$; delete $S_{k}$ from the feature vector group F to obtain a new feature vector group $F^{\prime }$.

Step 10. Assign the updated candidate feature sub-set $S^{\prime }$ to the candidate feature sub-set S; assign the updated feature vector group $F^{\prime }$ to *F*.

Step 11. Use the 10-fold cross-validation method to determine whether the classification accuracy of the candidate feature sub-set S on the category vector C is reduced when compared to the feature sub-set $S/S_{k}$ (representing the deletion of the element $S_{k}$ in the set S); if it is reduced, it means that the feature selection is complete; otherwise, k + 1 should be assigned to k, and the execution returns to step seven.

## 3. The Proposed Method

### 3.1. Data Pre-Processing

The primary function of data pre-processing was divided into two parts. The first step was to batch convert the obtained image files to a .bmp file format, which had nothing to do with hardware devices [23]. The format image did not use any compression, was lossless, and had accurate color. As a result, a .bmp file can take up a lot of storage space, but its application is still very broad. A typical .bmp file is divided into four sections: the bitmap file header data structure, the bitmap information data structure, the palette, and the bitmap data. Converting acquired images to a .bmp format makes image processing easier. The second step was to process the original .bmp image and segment it to obtain the image’s main portion. The segmentation method used in this paper was a graphics-based image-processing algorithm that performed erosion, median filtering, expansion, contour extraction, lung boundary contour extraction, flooding, and filling algorithms on the pneumonia image sequentially. In the first step, the basic feature region of the image was obtained after the erosion of the whole image. In the second step, the region of interest was obtained after the noise reduction and contour extraction. In the third step, the region of interest was filled with the source image; Figure 2 depicts the process and results of the image pre-processing.

### 3.2. YOLOv5 Deep Learning Model

The YOLO network divided the input image into S × S small grids, predicted in each grid separately, merged the results of each grid, and predicted two tasks in each grid: whether the grid contained features and the confidence in each included feature. It was no longer necessary to extract multiple ROIs for a single image, and the steps of NMS to delete ROIs with low confidence were saved, which greatly reduced calculation steps and time and improved the real-time performance. As shown in Figure 3. The combination of this network and mRMR not only improved the accuracy of the YOLOv5 feature extraction, but it also outperformed the combination of machine learning and CNN in real time. Multiple intermediate geometric models had to be predicted at the same time for the same image with multiple features. YOLO could produce more, whereas CNN had an error-prone process.

### 3.3. Proposed Method

The feature selection process of the deep-learning-based mRMR filtering small target feature algorithm is shown in the figure. All features of the fully connected layer in the last layer of the neural network were extracted, and their redundancy and relevance were evaluated. Finally, the recognition accuracy of the model was verified.

In this paper, mRMR was combined with the YOLO convolutional neural network, and the feature selection was divided into three stages, as shown in Figure 4.

In the first stage, the size of all images was scaled to 640 × 640, the images were divided into S × S small grids (S = 8 in this paper), and features were extracted for each grid. Finally, the network output features were B, the maximum number of overlapping cases, and C, the number of all cases in the data set.

In the second stage, mRMR was used to screen the features from the candidate features to remove the redundant features.

In the third stage, finally, the retained features were identified and classified, and then visualized to verify their accuracy of the retained features. In this paper, the nodule generated by pneumonia was used as a small target feature, which was observed in the visual image. The extracted features could reflect the robustness and accuracy of the algorithm well.

The image was divided into 8 × 8 small grids, each of which had 80 × 80 pixels. The network model had the fastest speed when identifying and classifying the 6 feature nodules.

## 4. Experiments and Simulations

In this paper, a YOLO neural network was trained by using a pneumonia data set, which contained 5849 samples and was publicly accessible. In this study, small features were extracted and nodules were visualized to ensure that the features selected using mRMR could effectively solve related problems. In this paper, the 5849 sample images were divided into 10 groups, and each group of samples was used as the test set to study the model in turn, and the remaining 9 groups were used as the training set to train the model. A ten-fold cross-validation method was used to obtain the best parameters of the model. The epochs of different groups were set to 50, 100, 150… 550. The Mini batch of each two groups was set to 16, 32, 64, 256, and 1024. The learning rate was set to 0.0001 for the first five groups and 0.00001 for the last five groups. Therefore, the ideal model parameters obtained in this paper are shown in Table 1. Under the following parameters, the model training time was longer, but the recognition accuracy was the highest, and the model convergence speed was faster.

Under the assumption that the model parameters remained constant, the mRMR algorithm and image enhancement technology were applied to the model. The table and Figure 5 showed that the mRMR algorithm had better feature extraction output results, which reduced the number of misjudgments and eliminated the output of redundant features to ensure better model results. In terms of the model training speed and accuracy, as shown in the figure, adding the mRMR algorithm had an effect on the model convergence speed, but had no effect on model accuracy. However, the YOLOv5 model could not be completely converged during the verification process, resulting in verification failure. The fluctuation of the function indicated that the YOLOv5 model had a limited ability to detect minor pneumonia characteristics. Although the loss remained, there was a significant improvement when compared to the original model, indicating that the accuracy of the feature recognition had improved.
(5)$$AC=\frac{TP+TN}{TP+TN+FP+FN}$$
(6)$$PR=\frac{TP}{TP+FP}$$
(7)$$RE=\frac{TP}{TP+FN}$$
(8)$$F_{1}=2\frac{PR\cdot RE}{PR+RE}=\frac{2TP}{FN+2TP+FP}$$
(9)$$FPR=\frac{FP}{FP+TN}$$

We compared the influence of the mRMR algorithm on the feature extraction. The original data set was used in the YOLOv5 model in the first step, and the mRMR algorithm was not used. To classify the pneumonia features and effectively output three values of TP, FN, FP, and TN, the end-to-end training method was used. In the second step, the original data set of the first step was used in the model, and the mRMR algorithm was used to classify and identify the learned features and effectively output three values of TP, FN, FP, and TN. In the third step, the image pre-processed data set was divided into a 70% training set and a 30% test set to train the model and effectively output the TP, FN, FP, and TN three values using the relevant 200 features selected with the mRMR method.

Table 2 was transformed into a bar graph, as shown in Figure 6. It can be seen intuitively that after processing the extracted features of the YOLO model with the mRMR algorithm, the AC, PR, and RE of the model could be effectively improved, and the misjudgment rate of features could be effectively reduced. After image pre-processing, the area of interest of the image was smaller, which further improved the AC, PR, and RE of the model and reduced the occurrence of a feature misjudgment.

Visualization tools were used in this paper to extract the features selected using the mRMR algorithm, and different features were color-labeled. As shown in Figure 7. The mRMR algorithm presented a good extraction of small target features, as shown in the visualized image. Using the pneumonia image as an example, because the morphological texture and shadow in the normal lung had significant influence on the feature extraction, this study employed a convolutional neural network to better extract the pneumonia image features and effectively reduce redundancy. The subsequent image processing was easier due to the identification and extraction of features.

## 5. Conclusions

The accuracy of the mRMR algorithm’s feature extraction was the focus of this paper. In this study, we used the existing YOLOv5 model as a feature extractor to run experiments on the features extracted using the network’s final connection layer. This paper employed the original network and the mRMR algorithm to extract the features extracted with the fully connected layer in order to validate the effectiveness of the experimental method. The experimental data showed that the YOLOv5 neural network, after applying the mRMR algorithm, could be used for small target features. To improve the extraction results at the same time, image enhancement technology was used in this paper to pre-process the image. The experimental data demonstrated that the pre-processed experimental image was superior for feature extraction. The results of this study showed that the mRMR algorithm could effectively reduce the extraction of redundant features, while improving classification efficiency and targeting feature extraction accuracy.

The algorithm could be used in different data sets in the future to study its universality, and the proposed method could be extended to a multi-task auxiliary diagnosis system that could be applied to medical diagnosis and industrial parts damage diagnosis.

## Figures and Tables

**Figure 1 micromachines-13-01765-f001:**
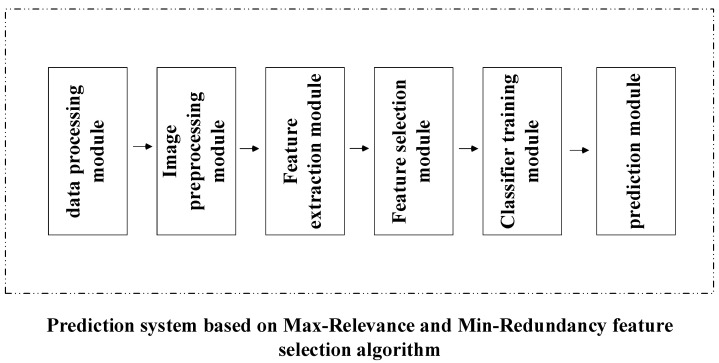
Computer-aided diagnosis.

**Figure 2 micromachines-13-01765-f002:**
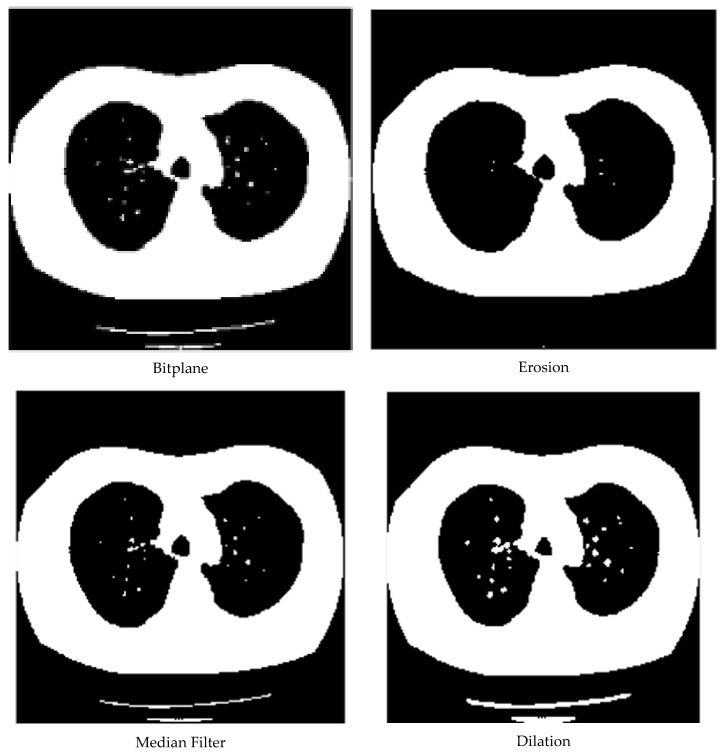
Image pre-processing process.

**Figure 3 micromachines-13-01765-f003:**
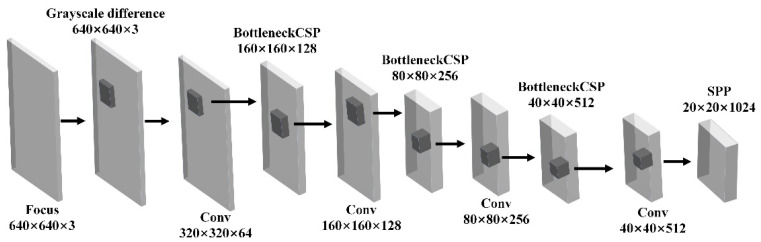
Feature extraction process of YOLOv5 model.

**Figure 4 micromachines-13-01765-f004:**
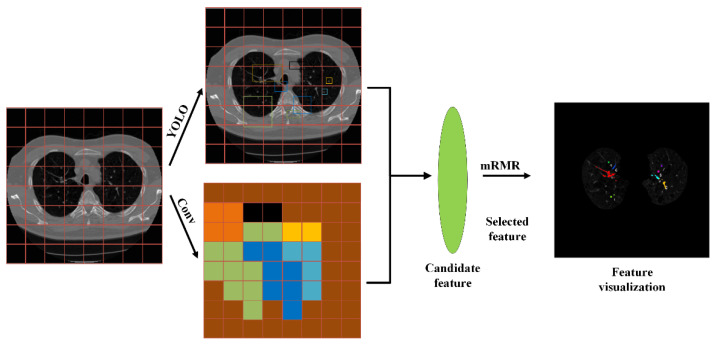
The method proposed in this paper.

**Figure 5 micromachines-13-01765-f005:**
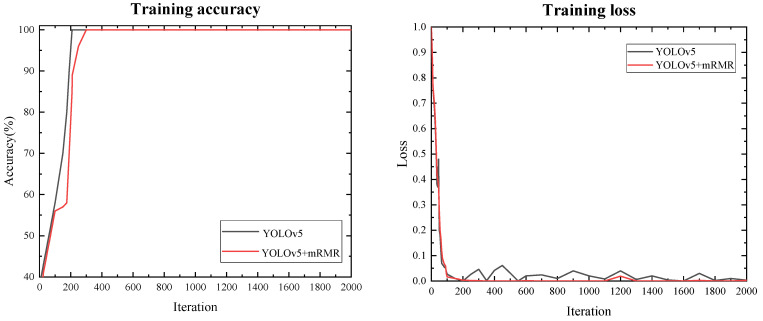
Pretrained graphs for both methods.

**Figure 6 micromachines-13-01765-f006:**
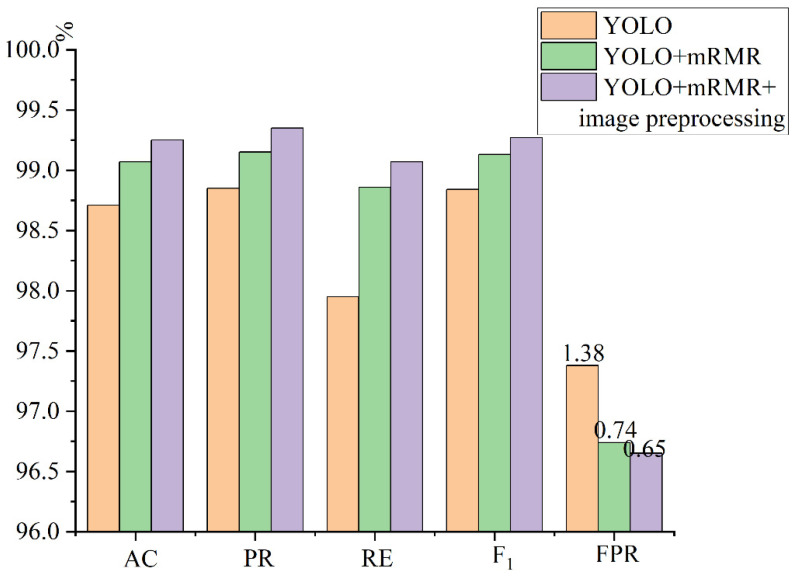
Comparative experimental results.

**Figure 7 micromachines-13-01765-f007:**
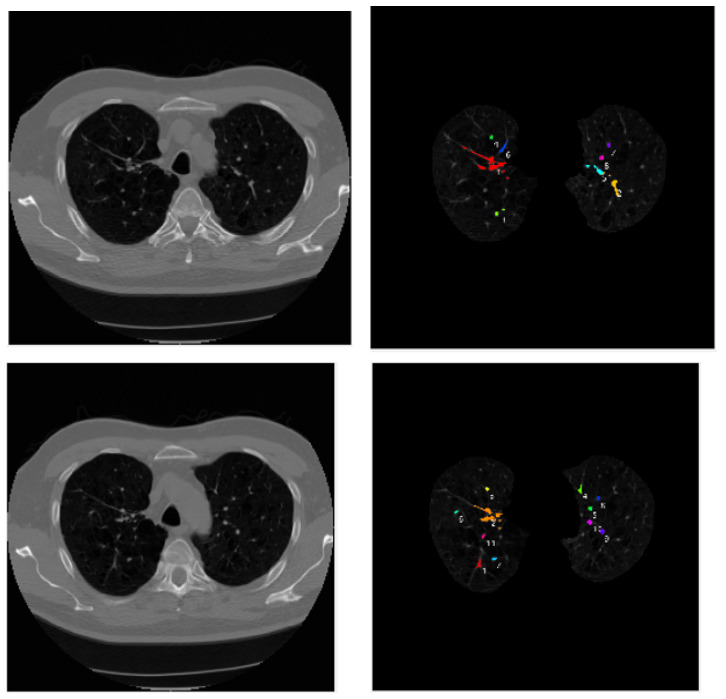
Pneumonia feature visualization results.

**Table 1 micromachines-13-01765-t001:** Deep learning model parameters.

Compile Software	Network Model	Image Size	Epochs	Mini Batch	Learning Rate
Pycharm2021	YOLOv5	640 × 640	300	16	0.0001

**Table 2 micromachines-13-01765-t002:** AC, PR, RE, F1, and FPR obtained with the application of the proposed method on the pneumonia data set.

Model	Features	AC (%)	PR (%)	RE (%)	F1 (%)	FPR (%)
YOLOv5	1024	98.71	98.85	97.95	98.84	1.38
YOLOv5 + mRMR	200	99.07	99.15	98.86	99.13	0.74
YOLOv5 + mRMR + image pre-processed	200	99.25	99.35	99.07	99.27	0.65

## Data Availability

The data used in this paper are public datasets. The public dataset can be downloaded at http://www.via.cornell.edu/lidc/ (accessed on 13 September 2022).

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
