# Peer review of "Deep Learning Based Feature Selection Algorithm for Small Targets Based on mRMR"

_micromachines, 2022, doi:10.3390/mi13101765_

Round 1

Reviewer 1 Report

Summary: The goal of the author is to use experimental findings  to show that the feature dimension may be efficiently reduced and small target features can be identified by applying the minimum redundancy maximum relevance method.   Here are the comments for the authors:
  • I do not understand the computer aided diagnosis section clearly. It looks like a pipeline based on your description but your figures looks like branches. The author should redraw the Figure to match the description given in lines 102 - 107. The author also mentioned that there is an unbreakable link between the modules, which means they are related.
  •  
  • Was SVM used for classification? Section 2.2 does mention the training of SVM classifier. However, section 3.2 shows that Yolo was used for the  Model training. So, why was training with SVM mentioned in section 2.2? What is the significance of mentioning SVM?
  •  
  • Figure 3 should be redrawn. It blurs out when zoomed out and font are too tiny when at normal view resolution.
  •  
  • In section 3.2, what is "SS" small grides in the first sentence?
  •  
  • Figure 5, x -axis is mislabeled.
  •  
  • Section 3.3, proposed method. This was so briefly explained. This is the whole essence of this paper, and it was described in just a paragraph? The authors stated "mRMR is combined with a YOLO convolutional neural network in this paper, and feature selection is divided into three stages". Each of these stages should be clearly described to accompany the provided Figure 4. For example, on of the questions an interested reader might have is this: what is the optimal value for the S small grid division? How do you determine this value?
  •  
  • Also, for the Deep learning model parameters in section 4, Table 1 was there a cross validation done to determine the best parameters?
  •  
  • Step 11 of section 2.3, the authors mentioned that they did ten cross validations but there is no mention of this throughout the paper or the results from this cross-validation.
  •  
  • There is no mention of source code or data availability used in this manuscript to test the credibility of the result and to support future development by other researchers.
  • Also, there is no need to add the confusion matrix. Table 2 is pretty much standard these days.

Author Response

Q1. I do not understand the computer aided diagnosis section clearly. It looks like a pipeline based on your description but your figures looks like branches. The author should redraw the Figure to match the description given in lines 102 - 107. The author also mentioned that there is an unbreakable link between the modules, which means they are related.

A1. Thank you for your valuable comments. It is true that the Computer Aided Diagnosis is related to each other. In this paper, the structure of the system is modified to be parallel because of the introduction of each branch.

Q2. Was SVM used for classification? Section 2.2 does mention the training of SVM classifier. However, section 3.2 shows that Yolo was used for the  Model training. So, why was training with SVM mentioned in section 2.2? What is the significance of mentioning SVM?

A2. Thank you for your advice. SVM classifier is one of the classic tools applied to feature classification, and R-CNN is to use SVM classifier for feature classification. Because SVM is used in the traditional computer-aided diagnosis system, this concept is wrongly introduced in the introduction of the theory. This paper actually uses multiple independent logistic classifiers for feature classification, which has been modified in the original paper.

Q3. Figure 3 should be redrawn. It blurs out when zoomed out and font are too tiny when at normal view resolution.

A3. Thank you for your valuable advice. The picture has been modified

Q4. In section 3.2, what is "SS" small grides in the first sentence?

A4. We are very sorry for the trouble caused to you due to our negligence. What we want to describe is S×S small grid, which has been modified.

Q5. Figure 5, x -axis is mislabeled.

A5. Thanks for your valuable advice. The picture has been redrawn.

Q6. Section 3.3, proposed method. This was so briefly explained. This is the whole essence of this paper, and it was described in just a paragraph? The authors stated "mRMR is combined with a YOLO convolutional neural network in this paper, and feature selection is divided into three stages". Each of these stages should be clearly described to accompany the provided Figure 4. For example, on of the questions an interested reader might have is this: what is the optimal value for the S small grid division? How do you determine this value?

A6. Thank you for your valuable advice. In this paper, the proposed method steps should be more detailed, especially the proposed feature extraction steps. The author's original intention is to introduce the proposed feature selection algorithm in detail in Section 2.3, and mainly explain the steps in the proposed method. Now this section has been modified to make the reader more clear and intuitive to see the work of each step in conjunction with Figure 4.

Q7. Also, for the Deep learning model parameters in section 4, Table 1 was there a cross validation done to determine the best parameters? Step 11 of section 2.3, the authors mentioned that they did ten cross validations but there is no mention of this throughout the paper or the results from this cross-validation.

A7. Thank you for your valuable advice, the author has modified it to address this problem. In this paper, the ten-fold cross-validation method is used not only in feature selection, but also in determining the parameters of the deep learning model to ensure the best training effect of the model.

Q8. There is no mention of source code or data availability used in this manuscript to test the credibility of the result and to support future development by other researchers.

Also, there is no need to add the confusion matrix. Table 2 is pretty much standard these days.

A8. Thank you for your valuable advice. The dataset used in this article, LIDC-IDRI, is linked to the public dataset below:

http://www.via.cornell.edu/lidc/

Thank you for your valuable advice, we have deleted the table.

Reviewer 2 Report

The English needs to be improved. There are a lot of incomplete sentences. Some examples are provided below
With respect to the proposed procedure. The reviewer cannot see the need for the preprocessing stage. The lungs are segmented out, and then YOLO is performed on the whole image. Why? 

Some specific comments:
Line 28: What is "Dimensionality Disaster Extraction"?

Lines 42-43: "Choose features with large differences; semi-supervised methods do not have a high labeling cost" -> Incomplete sentence

Line 69: "Remove any redundant features." -> Incomplete sentence

Equation (1) needs to be defined fully. What is I(xi;c)   Lines 100-101: To reduce the dimension of the extracted multi-dimensional 100 features. -> incomplete sentence   Line 131: What is an atlas?   Line 173: What do you mean by "Other than the image's depth, no other compression is used"   Figure 2: How do you get Extraction image after the Flood Filling image. This process has not been explained.   Line 201: Finally, the model is discovered. The precision is confirmed. -> Incomplete sentences   Figure 4: Why is the whole image divided into SxS grids, Why not only the segmented regions? Otherwise what is the purpose of the segmentation ie preprocessing stage?   Table 3 has not been explained in the text. What is mRMR+?

Author Response

Q1. With respect to the proposed procedure. The reviewer cannot see the need for the preprocessing stage. The lungs are segmented out, and then YOLO is performed on the whole image. Why?

A1. Thank you for your questions. We used image preprocessing as part of the comparison experiment to show that the preprocessed image can reduce the detection and recognition area of image features.

The model can be trained and tested with preprocessed images, which can improve the speed and accuracy of model recognition. In this paper, the requirements of the image preprocessing stage should be expressed more directly, and the experimental part has been modified to address this problem.

Q2. Some specific comments:

Line 28: What is "Dimensionality Disaster Extraction"?

A2. Thank you for your correction. Not "Dimensionality Disaster Extraction," but "the curse of dimensionality."

Q3. Lines 42-43: "Choose features with large differences; semi-supervised methods do not have a high labeling cost" -> Incomplete sentence

Line 69: "Remove any redundant features." -> Incomplete sentence

Lines 100-101: To reduce the dimension of the extracted multi-dimensional 100 features. -> incomplete sentence

Line 201: Finally, the model is discovered. The precision is confirmed. -> Incomplete sentences

A3. Thank you for your valuable suggestions, and the relevant sentences have been revised.

Q4. Equation (1) needs to be defined fully. What is I(xi;c)?

A4. Thank you for your valuable advice, we have a definition. This change makes it easier for the reader to understand the formula.

Q5. Line 173: What do you mean by "Other than the image's depth, no other compression is used"

A5. What I originally meant was bmp format image does not use any compression, lossless, and accurate color. And bmp format image can be modified image bit depth such as 8bits and 24bits. Under your reminder, we found that the expression was not appropriate, and it has been modified in this text.

Q6. Figure 2: How do you get Extraction image after the Flood Filling image. This process has not been explained.

A6. Thank you for your comments and we have covered the process in some detail. The explanation is as follows:

In the first step, the basic feature region of the image is obtained after the erosion of the whole image. In the second step, the region of interest is obtained after the noise reduction and contour extraction. In the third step, the region of interest is filled with the source image.

Q7. Figure 4: Why is the whole image divided into SxS grids, Why not only the segmented regions? Otherwise what is the purpose of the segmentation ie preprocessing stage?   Table 3 has not been explained in the text.

A7. Thank you for your advice. The original intention of the author is to use the preprocessed image as a comparison experiment, and image preprocessing can improve the speed of feature selection. Figure 4 is just an expression of the algorithm process. In the actual experiment process, the original image is first used in this paper, and then the preprocessed image is used for comparison experiment. In this paper, the preprocessed image is used to obtain the feature visualization image.

We have explained the Table as following:

“Table 2 is transformed into a bar graph as shown in Figure 6. It can be seen intui-tively that after processing the extracted features of YOLO model by mRMR algorithm, the AC, PR and RE of the model can be effectively improved, and the misjudgment rate of features can be effectively reduced. After image preprocessing, the area of interest of the image is smaller, which further improves the AC, PR, RE of the model and reduces the occurrence of feature misjudgment.”

问题 8.什么是移动通信技术+?

解答 8.感谢您的宝贵建议。对于由于我没有调整数字表格格式的疏忽而给您的评论带来的不便,我深表歉意。原始文章是 mRMR+ 图像预处理。

Round 2

Reviewer 1 Report

None. All the comments have been addressed.

Reviewer 2 Report

The authors have adressed all the issues raised .